# The Novel Association of Early Apoptotic Circulating Tumor Cells with Treatment Outcomes in Breast Cancer Patients

**DOI:** 10.3390/ijms23169475

**Published:** 2022-08-22

**Authors:** Evgeniya S. Grigoryeva, Liubov A. Tashireva, Vladimir V. Alifanov, Olga E. Savelieva, Sergey V. Vtorushin, Marina V. Zavyalova, Nadezhda V. Cherdyntseva, Vladimir M. Perelmuter

**Affiliations:** 1The Laboratory of Molecular Oncology and Immunology, Cancer Research Institute, Tomsk National Research Medical Center, Russian Academy of Sciences, 634050 Tomsk, Russia; 2The Department of General and Molecular Pathology, Cancer Research Institute, Tomsk National Research Medical Center, Russian Academy of Sciences, 634050 Tomsk, Russia

**Keywords:** breast cancer, metastasis, circulating tumor cells, apoptosis

## Abstract

Stemness and epithelial–mesenchymal plasticity are widely studied in the circulating tumor cells of breast cancer patients because the roles of both processes in tumor progression are well established. An important property that should be taken into account is the ability of CTCs to disseminate, particularly the viability and apoptotic states of circulating tumor cells (CTCs). Recent data demonstrate that apoptosis reversal promotes the formation of stem-like tumor cells with pronounced potential for dissemination. Our study focused on the association between different apoptotic states of CTCs with short- and long-term treatment outcomes. We evaluated the association of viable CTCs, CTCs with early features of apoptosis, and end-stage apoptosis/necrosis CTCs with clinicopathological parameters of breast cancer patients. We found that the proportion of circulating tumor cells with features of early apoptosis is a perspective prognosticator of metastasis-free survival, which also correlates with the neoadjuvant chemotherapy response in breast cancer patients. Moreover, we establish that apoptotic CTCs are associated with the poor response to neoadjuvant chemotherapy, and metastasis-free survival expressed at least two stemness markers, CD44 and CD133.

## 1. Background

Researchers’ dreams of using liquid biopsy for monitoring disease progression in solid tumors have not yet been materialized due to the objective difficulties. The main challenge lies in the high plasticity of tumor cells. Stemness and epithelial–mesenchymal plasticity in cancer are concepts that represent a cancer cell’s ability to utilize normal developmental programs to promote survival and expansion [1]. The non-binary (hybrid) states of these processes often complicate the analysis of the functional state of CTCs and makes the evaluation of their clinical significance challenging [2,3]. The group of experts [4] attempting to achieve consensus not only in understanding the mechanisms of epithelial–mesenchymal (EMT) and mesenchymal–epithelial transition (MET) but also to propose specific criteria for defining variants of “epithelial”, “mesenchymal” and “hybrid” phenotypes concluded that EMT/MET is not a linear sequential process. According to the authors, EMT is a process with many collaterals, each of which can go from one to another in different phases of the process. This phenomenon explains the high plasticity of EMT manifestations. It follows that today, there are no unambiguously interpreted protein molecules that would clearly identify different phenotypes during EMT. Moreover, the authors argue that the most correct and reliable way to identify different phenotypes may be the functional characteristics of cells. When attempting to characterize EMT in vivo, the ability to use these guidelines is limited. Given the above, the interpretation of EMT remains challenging. The pronounced plasticity of stemness manifestations, which primarily provides the ability of non-stem cells to transform into stem cells, as well as the functional features of stem cells that (co)express different markers of stemness and transcription factors are still not well understood. CD44 and CD133 are the most widely used markers in cancer stem cell research and are already used as therapeutic targets in cancers [5].

Circulating tumor cells (CTCs) with an epithelial/mesenchymal hybrid phenotype and stem cell-like signatures correlate with significantly reduced progression-free survival [6]. The reflection of tumor cells’ plasticity is the pronounced heterogeneity of CTCs, which is confirmed by a growing body of studies wherein CTCs are heterogeneous both within the individual patient and across study cohorts [7,8,9].

Although it is known that CTCs are associated with distant metastases, which in most cases are the cause of a cancer patient’s death, distinct markers that reflect properties that give advantages for metastasis formation have not yet been identified [10,11]. An important property that should be taken into account when studying the ability of CTCs to disseminate is viability and the state of cell death (apoptosis and necrosis). So, it was shown that the number of apoptotic CTCs (apoCTCs) in the peripheral blood of cancer patients is associated with the prognosis of the disease [12,13]. It should be noted that the detection of apoCTCs in vivo is a difficult task since the lifespan of CTCs in the bloodstream is very short. Wei X. et al. showed that the half-life of CTCs in circulation is approximately 10 min [14], and a study conducted by Meng et al. showed that the half-life of CTCs after surgical removal of primary breast tumors is 1–2 h [15].

There are several methods for the detection of apoCTCs both in vivo and in vitro. Molecular imaging tools, such as single photon emission computed tomography (SPECT), positron emission tomography (PET), magnetic resonance imaging (MRI) and optical imaging, could be used to study apoptosis in vivo [16].

Apoptosis analysis in vitro detects and evaluates the cellular events associated with programmed cell death, including caspase activation, cell surface exposure of phosphatidylserine and DNA fragmentation. Kallergi G et al., using a method based on the identification of activated caspases, stated that the median percentage of apoCTCs per patient was 53.6% in patients with early metastatic disease and 0% in patients with metastatic disease [12]. At the same time, another method of apoptosis detection with M30 antibody conducted by the authors revealed a similar pattern but a different proportion of apoCTCs. The median percentage of apoCTCs per patient was 80% in patients with early metastatic disease and 15% in patients with metastatic disease. The authors explained the discrepancy by the fact that each assay evaluated different steps of the apoptotic process: the first one characterized the activation of caspases, whereas the second identified a neo-epitope of cytokeratin 18 created after caspase cleavage [12]. Annexin-based assay with fluorescent dyes provides reproducible and relatively fast detection of early apoptosis. This method utilizes fluorescently conjugated annexin-V, a small 35 kDa protein, for the labeling of phosphatidylserine on the outer surface of the plasma membrane of cells undergoing early apoptosis [16]. 7AAD dye is commonly used for the exclusion of nonviable cells in flow cytometric analysis [17]. Using 7AAD and annexin V allows cells to be distinguished between viable, early and late/dead apoptotic cells [18]. 

Another promising area for liquid biopsy is using it as a tool for monitoring antitumor treatment efficacy [19,20]. However, the results of a meta-analysis indicate that there is no association between the level of CTCs and pathological complete response (pCR) after neoadjuvant chemotherapy in breast cancer patients [21]. A later meta-analysis showed that the level of CTCs was associated with the overall survival and distant disease-free survival, but not with the neoadjuvant chemotherapy response [22].

The aim of our study was to investigate the association of CTCs in different apoptotic states with short- and long-term treatment outcomes. We also evaluated exactly which features of stemness and EMT were essential for apoCTCs associated with decreased metastasis free-survival and poor neoadjuvant chemotherapy response.

## 2. Results

Our study included 58 breast cancer patients with CTCs; the full clinicopathological parameters of the patients are presented in Table 1. CTCs were extremely rare cells and amounted on average to 0.01–0.02% of all white blood cells in CTC-positive patients.

To investigate the baseline of apoptosis in CTCs, blood samples were drawn prior to neoadjuvant chemotherapy (group of untreated patients) or surgery (group of patients treated with neoadjuvant chemotherapy). Given that apoptosis time frames are more continuous compared to the CTC lifespan, we suppose that the process is most likely initiated in the primary tumor [23,24]. Therefore, the effects detected by liquid biopsy could be extrapolated to the primary tumor. 

### 2.1. apoCTCs, Non-apoCTCs and End-Stage apoCTCs in Peripheral Blood of BC Patients

We evaluated frequencies of CTCs in different states of apoptosis in untreated patients and patients treated with neoadjuvant chemotherapy (NAC). apoCTCs (34/40) were detected in 85% of untreated patients and in 72% (13/18) of patients treated with NAC, while end-stage apoCTCs were found rarely, only in 10% (4/40) of untreated patients and in 6% (1/18) of patients treated with NAC. Non-apoCTCs were observed in 100% of patients independently of NAC treatment.

Figure 1 shows percentages of apoCTCs, non-apoCTCs and end-stage apoCTCs of distinct phenotypes: stem and non-stem (CD44+CD24−/CD44−CD24−) with simultaneous positive/negative expression of N-cadherin. The percentage of non-apoCTCs in untreated patients was 39.29% (21.83–62.50%), while 8.14% (2.61–16.67%) of cells were apoCTCs, and 0.00% (0.00–0.00%) were end-stage apoCTCs (Figure 1A). Percentages of CTCs in the different states of apoptosis in patients treated by NAC were 52.50% (34.25–72.22%), 2.08% (0.00–4.90%) and 0.00% (0.00–0.00%), respectively (Figure 1B). 

The percentages of non-apoCTCs, apoCTCs and end-stage apoCTCs in each breast cancer patient are presented in Appendix A. Figure 1 contains the percentages of apoCTCs, non-apoCTCs and end-stage apoCTCs of distinct phenotypes: stem and non-stem (CD44+CD24−/CD44−CD24−) with simultaneous positive/negative expression of N-cadherin. CD44+CD24+ and CD44−CD24+ CTCs were not studied. For this reason, the variety of all studied phenotypes does not represent 100% of all possible phenotypes.

In the total pool of CTCs in breast cancer patients, the smallest proportion of CTCs was represented by end-stage apoCTCs, regardless of the absence (A) or presence (B) of NAC (Figure 1). The proportion of apoCTCs in breast cancer patients was significantly lower than non-apoCTCs and higher than end-stage apoCTCs.

### 2.2. apoCTCs, Non-apoCTCs and End-Stage apoCTCs in Breast Cancer Patients in Relation to Clinicopathological Characteristics

We compared the proportions of apoCTCs, non-apoCTCs and end-stage apoCTCs in breast cancer patients (n = 58) with clinicopathological parameters such as molecular subtype of primary tumor, tumor size (T), lymph node involvement (N), tumor grade and occurrence of distant metastasis in follow-up period (data not shown). We evaluated CTC proportions separately for patients treated and not treated with NAC. No differences in apoCTC proportions in untreated patients and patients treated with NAC with different molecular subtypes were detected (Figure 2). Jansson et al. also found no correlation between apoCTC and molecular subtype when assessing the prognostic value of apoCTCs in metastatic breast cancer patients [13].

Significant differences were observed only in groups with different tumor sizes (T) (Figure 3A,B). So, the proportion of non-apoCTCs was significantly higher (59.68% (29.37–93.04%)) in untreated T1 patients compared to T2 patients (32.67% (16.06–61.11%)) (*p* = 0.0019) (Figure 3A).

While in patients treated with NAC, there was a controversial pattern, the proportion of apoCTCs was higher in T2 (57.14% (32.57–84.58%)) compared to T1 (35.47% (10.27–66.67%)) breast cancer patients (*p* = 0.0410) (Figure 3B).

### 2.3. Prognostic Significance of apoCTCs and Non-apoCTCs for Metastasis-Free Survival

Although there were no differences in the proportion of apoCTCs in patients with and without distant metastasis in the follow-up period, we decided to evaluate the prognostic significance of the proportion of such subpopulations for metastatic-free survival. It was impossible to assess the prognostic value of the end-stage apoCTC population due to the extremely small proportion of these cells among the CTCs. We evaluated the cut-off for proportions of non-apoCTCs and apoCTCs for predicting distant metastasis in breast cancer patients (n = 58) (Figure 4).

We assessed outcomes at a median follow-up of 22 months. No correlation of total CTC count with metastasis-free survival was found (*p* = 0.4754). Furthermore, we assessed metastasis-free survival in patients with CTC percentages under and above the established cut-off. In patients with a proportion of apoCTCs above the cut-off, the metastasis-free survival rate was significantly lower, at 33.3%, while in patients with a proportion below the cut-off, the metastasis-free survival rate was 69.3% (*p* < 0.0001) (Figure 4A). Metastasis-free survival for patients with proportions of non-apoCTCs above and below the cut-off were 90% and 61.3%, respectively (*p*-value was insignificant).

Next, we focused on apoCTCs, as the differences in metastasis-free survival were significant. The proportion of apoCTCs above the cut-off was an independent prognostic marker of poor metastasis-free survival in breast cancer patients (Table 2).

### 2.4. Association of apoCTCs Proportion with NAC Response

We evaluated proportion of non-apoCTCs, apoCTCs and end-stage apoCTCs in patients with different responses to the NAC (n = 18). Standardized response evaluation after NAC can be assessed by the residual cancer burden (RCB) index, which is quantified based on the primary tumor area, the percentage of the tumor area that is invasive cancer and the extent of lymph node involvement [25]. RCB0 corresponds to a pathological complete response (pCR) to NAC, and it is defined as the complete disappearance of the tumor and lymph node at the time of surgery. Non-pCR tumors can be categorized into one of the following three classes: RCB-I (minimal burden), RCB-II (moderate burden) or RCB-III (extensive burden). We merged patients with pCR and RCB-I, as they demonstrated favorable response to the NAC.

The apoCTC proportion was significantly higher in patients with RCB-III compared to patients with RCB-II (*p* = 0.047). The apoCTC percentages amounted to 14.27% (3.28–25.96%) and 2.08% (0.139–3.61%) (Me (Q1–Q3) in patients with RCB-III and RCB-II, respectively (Figure 5).

The obtained data suggest that the apoCTC proportion correlates with unfavorable response to neoadjuvant chemotherapy in breast cancer patients.

### 2.5. Phenotypic Characteristics of Cells Associated with Response Neoadjuvant Chemotherapy

Among all non-apoCTCs, apoCTCs and end-stage apoCTCs, we evaluated the proportion of cells with stemness features (CD44+CD24−) and features of late EMT (N-cadherin+) (Figure 6). The increased apoCTC and decreased non-apoCTC proportions in patients with RCB-III were characteristic of cells with stem features. Thus, stem CTCs, but not CTCs with EMT features, reproduced the pattern we discovered earlier (Figure 5), and consequently, it was the stem phenotype that was characteristic of apoCTCs associated with response to neoadjuvant chemotherapy. Patients with RCB-III had significantly increased proportions of stem apoCTCs and decreased proportions of stem non-apoCTCs compared to patients with RCB-II (Figure 6).

We clarified the phenotype associated with the response to neoadjuvant chemotherapy by epithelial markers and additional markers of stemness. It turned out that EpCAM+CK7− cells with features of EMT predominated among CD44+CD24− stem apoCTCs. The proportion of EpCAM+CK7+ and EpCAM-CK7+ cells did not differ by N-cadherin expression (Figure 7).

Given that CD44 is not the only marker of stemness, and that it is more common in mesenchymal stem cells [26], we assessed stemness of apoCTCs considering two other markers, namely, CD133 and ALDH1A1. It turned out that most apoCTCs associated with the poor response to neoadjuvant chemotherapy and metastasis-free survival expressed at least two stemness markers, CD44 and CD133, and there was often additional expression of ALDH1 (Figure 8).

## 3. Discussion

The total pool of CTCs is represented by a heterogeneous population of cells, while only a minority of CTCs have the potential to generate metastases [27]. Most detached tumor cells will die in the bloodstream because losing anchorage induces anoikis, a form of cell death caused by apoptosis [28]. Currently, there is no universal method for detecting apoptotic cells; rather, a group of different methods is used for this goal, namely, electron microscopy, genomic methods, spectroscopic techniques, caspase activity assay and flow cytometry [29]. Each of these methods has their own advantages and limitations. The main advantage of flow cytometry, despite the impossibility of morphological assessment of cells, is the analysis of huge cell count. Currently, there are many commercially available reagents for assessing apoptosis by flow cytometry, namely, SYTO [30], annexin-V, M30 and others [31]. The most common and reproducible approach for apoptosis evaluation is the method based on labeling cells with Annexin V fluorochrome conjugate and 7-AAD [12,16].

The CellSearch system cut-off of five or more CTCs per 7.5 mL of blood has been validated as a predictor of poor prognosis in breast cancer patients. However, this approach does not take into account the contribution of cells that play a role in the spread of viable tumor deposits and those that are biologically inert [32]. The study of Deutsch T.M. et al. demonstrated that the same cut-off, 5 apoCTCs/7.5 mL, is a predictor of unfavorable prognosis for metastatic breast cancer patients [33]. In our study, the proportion, but not the number, of apoCTCs above the cut-off predicted the decreased metastatic-free survival. The cut-off was established as 33.55% apoCTCs. This may indicate that it is necessary to consider not the total count of CTCs, but the ratio of CTCs associated with adverse disease outcomes. That is, even with a low level of CTCs, their ratio may indicate the risk of unfavorable outcomes. Conversely, a high level of CTCs with a low proportion of apoCTCs can stratify patients with a favorable course of the disease. This will allow a more accurate prognosis and administration of adequate therapy.

We also evaluated apoCTC proportions in patients with different molecular subtypes of primary tumor, tumor size, lymph node status and tumor grade. In our study, there were no differences in CTC proportions in breast cancer patients in relation to clinicopathological parameters, except tumor size. This is probably due to the division of all CTCs according to their apoptotic state. Interestingly, the apoCTC percentage was significantly different in patients with different tumor sizes. Particularly, in patients with larger tumors (>5 cm), the proportion of apoCTCs was significantly higher. These data correspond to another study that demonstrated an increased number of CTCs in patients with large tumors [34].

We evaluated long-term outcomes after 2 years of follow-up. All patients were alive, but four patients had developed distant metastasis, so the endpoint of our study was metastasis-free survival. It is well-known that CTCs constitute a powerful prognostic marker in cancer patients [35,36,37]. In our study, we evaluated the prognostic significance of distinct CTC populations according to their apoptotic states. It turned out that the proportion of apoCTCs was significant for predicting metastasis-free survival. Furthermore, we have shown the independent significance of this parameter from the conventional predictive factors in breast cancer patients.

It is commonly assumed that the leading role in metastasis belongs to cells without features of apoptosis, which obtain the possibility to proliferate at the site of future metastasis. However, our results suggest that an increased proportion of apoCTCs is associated with poor metastasis-free survival. Probably, the explanation of such a phenomenon lies in the reversibility of early apoptosis. Furthermore, Xu Y. et al. (2018) demonstrated that apoptosis reversal promotes the formation of cancer cells with pronounced features of stemness [38]. Immunophenotyping revealed increased percentages of CD44+CD24− cells in the reversed breast cancer cell population. The authors report that human mammary carcinoma cells induced to undergo apoptosis could recover with increased tumorigenicity, both in vitro and in vivo, and induced lymph node metastasis. Our data indirectly confirm such an observation, because of the minimal presence of two stemness markers—CD44 and CD133—on apoCTCs which are associated with poor prognosis in breast cancer patients. Furthermore, apoCTCs were characterized by N-cadherin expression, which indicates late epithelial–mesenchymal transition [39]. Thus, apoCTCs demonstrate significant plasticity, which allows them to detach from the primary tumor and not die by apoptosis, while acquiring significant potency for further metastasis.

Moreover, these properties correlated with the response of the primary tumor to NAC. We suppose that most likely, the initiation of apoptosis takes place in primary tumors, because the time frames of apoptosis are wider than the mean lifespan of CTCs [14,24]. We may assume that the apoptosis status of CTCs could correlate with the cancer cells in primary tumors. It is known that the mechanism of action of chemotherapy drugs is mainly associated with the induction of apoptosis in tumor cells [40,41]. Probably, a high baseline of apoptosis in cells of the primary tumor deprives the drugs of the point of application and they are ineffective. In addition, the observed association of apoCTCs with the response to neoadjuvant chemotherapy can be explained by the pronounced stemness features of apoCTCs. It is well known that chemoresistance may be associated with the stem state of cells [42]. Chemoresistance of cancer stem cells may be provided by several factors, namely, by the activation of drug efflux mechanisms (ABC family transporters) and the multidrug resistance P-glycoprotein (P-gp), and by the overexpression of DNA repair mechanisms, including homologous recombination, non-homologous end joining and base excision repair through increased poly (ADP-ribose) polymerase 1 (PARP1) activity [42]. 

A major limitation of the present study includes our inability to express data in absolute values because we did not evaluate the entire spectrum of possible phenotypes, as we described in the Materials and Methods section. However, it was this approach that made it possible to identify the proportion of cells in early apoptosis, clinically relevant for assessing treatment outcomes. The group of patients with metastases was small, but even in such a small cohort, it was possible to find an association with metastatic-free survival. This gives us reason to believe that validation on a large cohort of breast cancer patients in follow-up will confirm the found associations. One more limitation is linked with the absence of CTCs in the peripheral blood of a significant group of breast cancer patients. Probably, in such cases, it is necessary to determine the presence of CTC repeatedly.

apoCTCs are a promising object of research for the development of a personalized approach to the treatment of malignant neoplasms, as their proportion is an informative predictor of metastatic-free survival. At the same time, stemness and EMT features are obligatory properties of apoCTCs, once again highlighting the importance of these properties in tumor progression.

## 4. Materials and Methods

### 4.1. Patients

The prospective study included 58 patients with invasive breast carcinoma of no special type (IC NST) T1-4N0-3M0, admitted for treatment to the Cancer Research Institute, Tomsk National Research Medical Center. The study was approved by the Local Committee for Medical Ethics of the institute (17 June 2016, approval No. 8), and informed consent was obtained from all patients prior to analysis. Venous ethylenediaminetetraacetic acid (EDTA) blood samples were taken before surgery and neoadjuvant chemotherapy. Patients were treated according to ESMO Clinical Practice Guidelines [43]. Evaluation of response to neoadjuvant chemotherapy was performed using RCB score and achievement of pCR [44].

### 4.2. Blood Specimen Collection and Processing for CTCs Immunophenotyping

Blood samples were collected in EDTA pre-coated 9 mL tubes, then incubated at 37 °C for 1.5 h. White blood cells were aspirated from the thin white layer between plasma and red blood cells after their sedimentation. The obtained cell concentrate was washed in 2 mL Cell Wash buffer (BD Biosciences, San Jose, CA, USA) by centrifugation at 800× *g* for 15 min.

### 4.3. Flow Cytometry

Surface markers (CD45, EpCAM(CD326), CD44, CD24, CD133 N-cadherin(CD325)), as well as Annexin-V and 7-AAD were stained in the first step; intracellular staining was performed in the second step. Samples were incubated at RT for 10 min with 5 μL of Fc Receptor Blocking Solution (Human TruStain FcX, Sony Biotechnology, San Jose, CA, USA). Next, monoclonal antibodies were added and incubated at RT for 20 min: APC-Cy7-anti-CD45 (clone HI30, IgG1, Sony Biotechnology, San Jose, CA, USA), BV 650-anti-CD326 (clone 9C4, IgG2b, Sony Biotechnology, San Jose, CA, USA), PE-Cy7-anti-N-cadherin (clone 8C11, IgG1, Sony Biotechnology, San Jose, CA, USA), BV 510-anti-CD44 (clone G44-26, IgG2b, BD Biosciences, San Jose, CA, USA), PerCP-Cy5.5-anti-CD24 (clone ML5, IgG2a, Sony Biotechnology, San Jose, CA, USA), BV 786-anti-CD133 (clone 293C3, IgG2b, BD Biosciences, San Jose, CA, USA), FITC-conjugated Annexin V (Sony Biotechnology, USA) and 7-AAD (Sony Biotechnology, San Jose, CA, USA). The unstained control and antibody quality control were performed. The appropriate isotype antibodies were added to the isotype control sample at the same concentration. After incubation, red blood cells were lysed by 250 μL OptiLyse C buffer (Beckman Coulter, Brea, CA, USA) at RT for 10 min in the dark and washed in 1 mL Cell Wash buffer (BD Biosciences, San Jose, CA, USA) at 800× *g* for 6 min.

For intracellular staining, cells were permeabilized by 250 μL BD Cytofix/Cytoperm (BD Biosciences, San Jose, CA, USA) at 4 °C for 30 min in the dark and washed twice in 1 mL BD Perm/Wash buffer (BD Biosciences, San Jose, CA, USA) at 800× *g* for 6 min. After, samples were diluted in 50 μL BD Perm/Wash buffer (BD Biosciences, San Jose, CA, USA) and incubated at 4 °C for 10 min in the dark with 5 μL of Fc Receptor Blocking Solution (Human TruStain FcX, Sony Biotechnology, San Jose, CA, USA). Next, monoclonal antibodies were added and incubated at 4 °C for 20 min: AF647-anti-CK7/8 (clone CAM5.2, Mouse IgG2a, BD Biosciences, San Jose, CA, USA), BV 650-anti-CD326 (clone 9C4, IgG2b, Sony Biotechnology, San Jose, CA, USA) and PE-anti-ALDH1A1 (clone 03, IgG1, Sino Biological, Beijing, China). The appropriate isotype control antibodies at the same concentration were added to the control sample. After incubation, samples were washed in 1 mL Cell Wash buffer (BD Biosciences, San Jose, CA, USA) at 800× *g* for 6 min. After, samples were diluted in 100 μL stain buffer (Sony Biotechnology, San Jose, CA, USA). Compensation beads (VersaComp Antibody Capture Bead kit, Beckman Coulter, Brea, CA, USA) were used for compensation control. The immunofluorescence was analyzed on the Novocyte 3000 (ACEA Biosciences, San Diego, CA, USA). 

The gating strategy was as follows: using forward (FSC) and side scatter (SSC) gates, debris was discriminated; doublets were also discriminated by plotting FSC area vs FSC height. Subsequent analysis included only CD45-negative cells, CTCs gated using a quadrant-based scheme using EpCAM and cytokeratin 7/8 to distinguish among the three CTCs subsets: EpCAM+CK7/8−; EpCAM-CK7/8+; EpCAM-CK7/8−. Next, the following phenotypes were evaluated in each of the listed subpopulations: stem and non-stem (CD44+CD24−/CD44−CD24−); with and without EMT features (N-cadherin+/N-cadherin−).

Finally, we characterized each subpopulation with a detailed phenotype by the expression of additional markers of stemness, ALDH1A1 and CD133, with simultaneous labeling by Annexin V and 7-AAD. Annexin V−/7-AAD− cells were considered as CTCs without features of apoptosis (non-apoCTCs), Annexin V+/7-AAD− cells as early apoptotic CTCs (apoCTCs) and Annexin V+/7-AAD+ cells as end-stage apoptotic or necrotic CTCs (end-stage apoCTCs). The obtained values were expressed as a percentage of all studied CTC phenotypes. 

### 4.4. Statistical Analysis

The data were analyzed using GraphPad Prism 9 (GraphPad Software, San Diego, CA, USA). The Mann–Whitney test was used to compare differences between independent groups; for the dependent variables, the Wilcoxon test was used. Metastasis-free survival was assessed with univariate and multivariate Cox regression models, and resulted in hazard ratios (HRs). This model adjusts for menopausal status, Ki-67 expression, tumor grade, stage T, N, quantity of apoCTCs and molecular subtype of the primary tumor. Metastasis-free survival was calculated by the Kaplan–Meier method, and differences in survival curves among the groups were evaluated by the log rank test. Cox regression analysis was performed to assess the prediction power of the apoCTC quantity in metastasis-free survival. *p* < 0.05 was considered statistically significant.

## Figures and Tables

**Figure 1 ijms-23-09475-f001:**
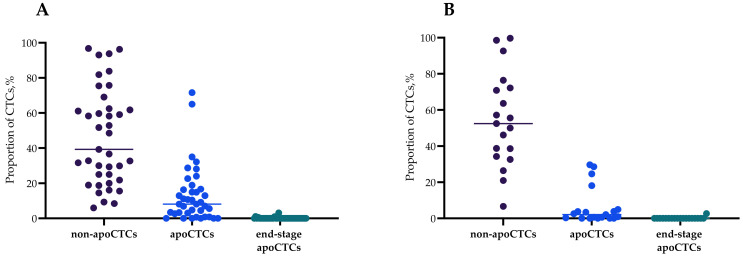
Percentage of non-apoCTCs, apoCTCs and end-stage apoCTCs in untreated breast cancer patients (**A**) and breast cancer patients treated with neoadjuvant chemotherapy breast cancer patients (**B**), Me(Q1–Q3).

**Figure 2 ijms-23-09475-f002:**
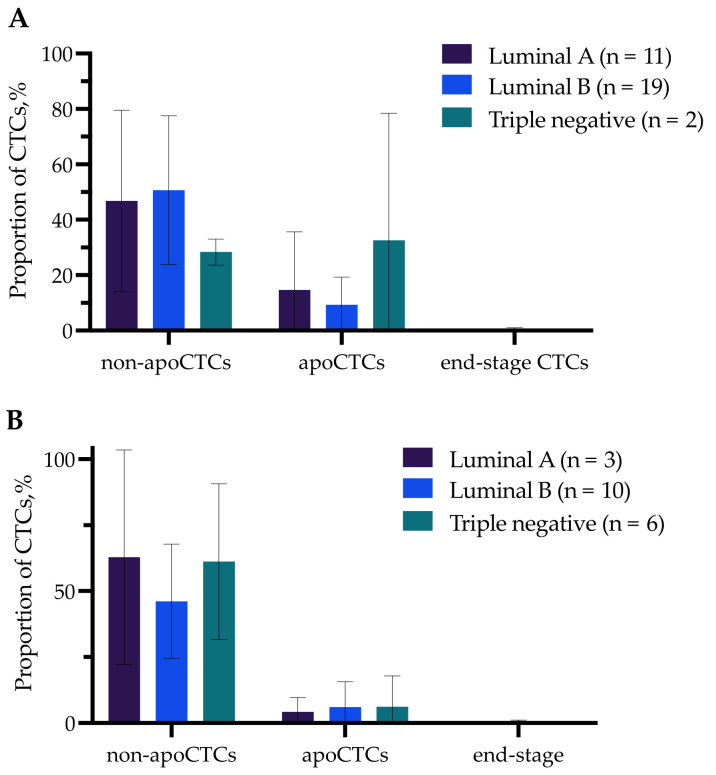
Percentage of non-apoCTCs, apoCTCs and end-stage apoCTCs in untreated breast cancer patients (**A**) and breast cancer patients treated with NAC (**B**) with different molecular subtypes.

**Figure 3 ijms-23-09475-f003:**
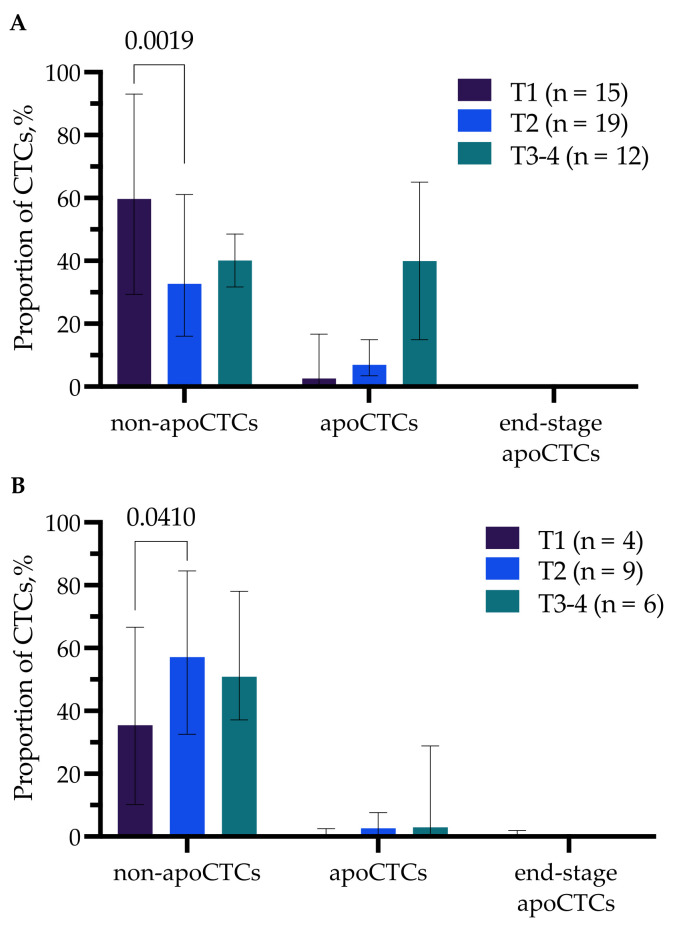
Percentage of non-apoCTCs, apoCTCs and end-stage apoCTCs in untreated breast cancer patients (**A**) and breast cancer patients treated with NAC (**B**) with different tumor sizes (T).

**Figure 4 ijms-23-09475-f004:**
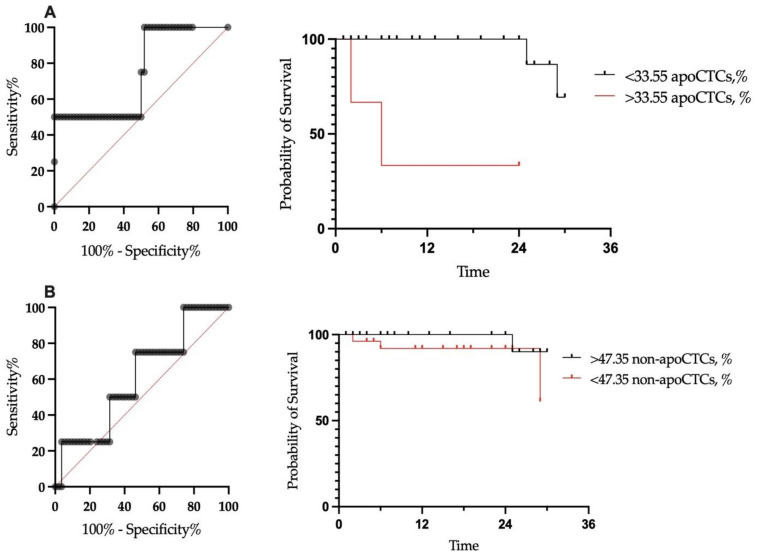
Metastatic-free survival of breast cancer patients in relation to proportion of apoCTCs (**A**) and non-apoCTCs (**B**).

**Figure 5 ijms-23-09475-f005:**
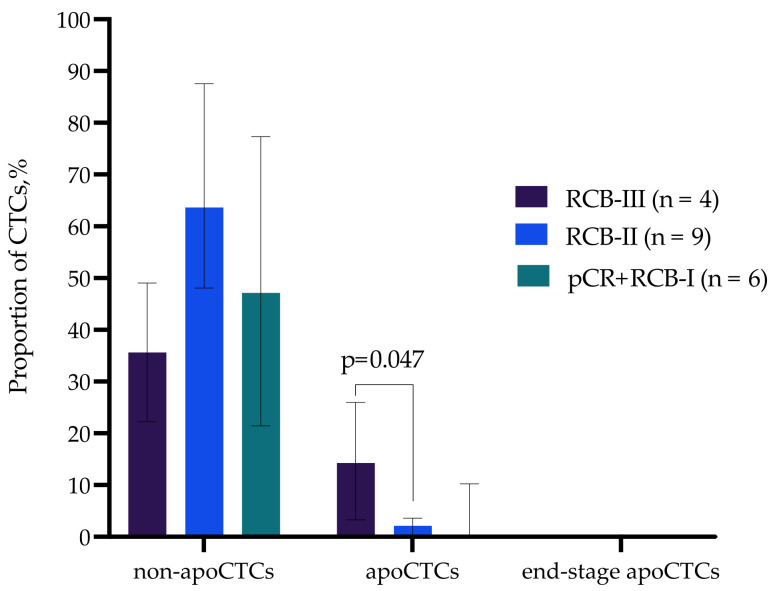
Proportion of non-apoCTCs, apoCTCs and end-stage apoCTCs in breast cancer patients with different responses to neoadjuvant chemotherapy.

**Figure 6 ijms-23-09475-f006:**
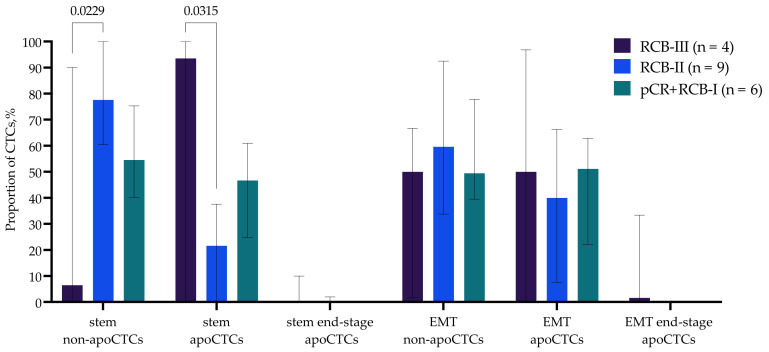
Proportion of CD44+CD24− CTCs and N-cadh+ CTCs with EMT features in breast cancer patients with different responses to neoadjuvant chemotherapy.

**Figure 7 ijms-23-09475-f007:**
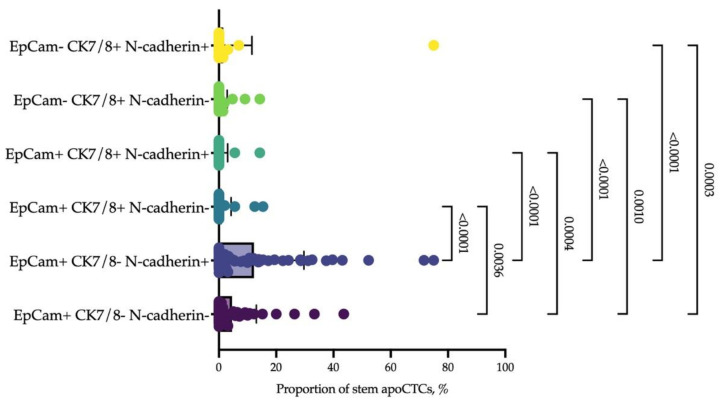
Epithelial and mesenchymal markers of stem apoCTCs. Each point represents the proportion of CD44+CD24− apoCTCs (considered as 100%) with epithelial features, EpCAM and cytokeratin 7/8 (CK7/8), and mesenchymal features (N-cadherin) in each breast cancer patient.

**Figure 8 ijms-23-09475-f008:**
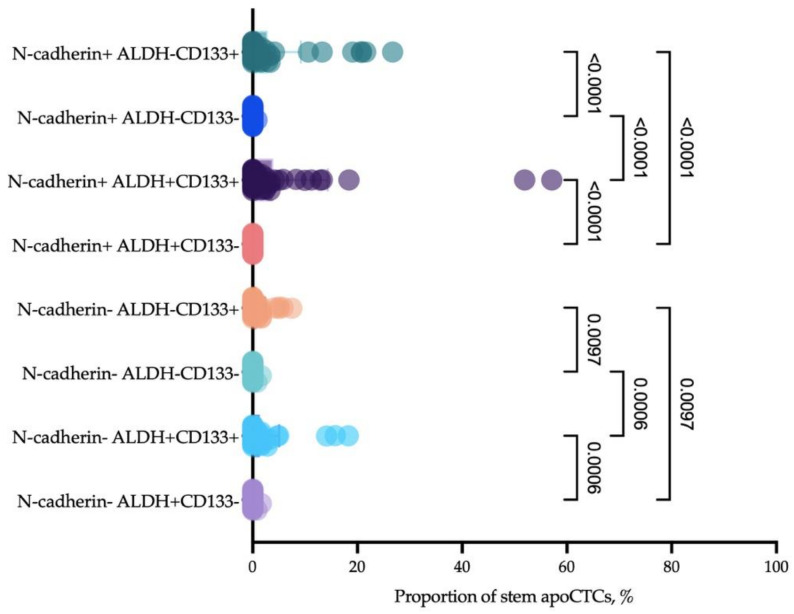
Proportion of CD44+CD24− apoCTCs (considered as 100%) that expressed mesenchymal features (N-cadherin) and additional features of stemness (CD133 and ALDH1).

**Table 1 ijms-23-09475-t001:** Clinicopathological characteristics of patients.

Parameter		Frequency, % (n)
Age	<35	5.17% (3/58)
	35–50	32.76% (19/58)
	>50	62.07% (36/58)
Menstrual function	Premenopausal	34.48% (20/58)
	Postmenopausal	65.52% (38/58)
Tumor size (T)	1	32.76% (19/58)
	2	53.45% (31/58)
	3	5.17% (3/58)
	4	8.62% (5/58)
Stage	I	20.69% (12/58)
	IIA	34.48% (20/58)
	IIB	25.86% (15/58)
	IIIA	3.45% (2/58)
	IIIB	12.07% (7/58)
	IIIC	3.45% (2/58)
Molecular subtype	Luminal A	25.86% (15/58)
	Luminal B	56.90% (33/58)
	Triple negative	15.52% (9/58)
	HER2 positive	1.72% (1/58)
Tumor grade	1	17.24% (10/58)
	2	67.24% (39/58)
	3	15.52% (9/58)
Estrogen receptor α	positive	82.76% (48/58)
	negative	17.24% (10/58)
Progesterone receptor	positive	70.69% (41/58)
	negative	29.31% (17/58)
HER2	positive	25.86% (15/58)
	negative	74.14% (43/58)
Ki67 expression	<20%	34.48% (20/58)
	>20%	65.52% (38/58)
Lymph node metastasis	Yes	37.93% (22/58)
	No	62.07% (36/58)
Distant metastasis	Yes	6.90% (4/58)
	No	87.93% (51/58)
	No data	5.17% (3/58)
Neoadjuvant chemotherapy	Yes	31.03% (18/58)
	No	68,97% (40/58)

**Table 2 ijms-23-09475-t002:** The univariate and multivariate Cox regression analyses of metastasis-free survival rate for breast cancer patients.

		Univariate		Multivariate	
		OR (95%CI)	*p*-Value	OR (95%CI)	*p*-Value
apoCTCs	<33.55%	ref		ref	
	>33.55%	27.62 (1.74–338.20)	0.0001	13.73 (1.34–182.33)	0.0002
Neoadjuvant chemotherapy	No	ref		ref	
	Yes	0.31 (0.03–3.44)	0.3446	0.48 (0.04–5.93)	0.4532
Menopausal	No	ref		ref	
	Yes	1.47 (0.29–23.98)	0.2830	0.33 (0.03–6.19)	0.4011
Molecular type	Lum	ref		ref	
	TN	0.32 (0.02–5.28)	0.4383	0.27 (0.02–5.617)	0.3998
Ki67	<20	ref		ref	
	>20	0.94 (0.08–10.49)	0.9651	1.04 (0.12–7.13)	0.7782
Stage	I–II	ref		ref	
	III–IV	1.11 (0.19–2.42)	0.9995	1.15 (0.20–6.33)	0.7811
Grade	1	ref		ref	
	2	0.18 (0.01–3.15)	0.1712	0.28 (0.01–2.88)	0.3190
	3	0.83 (0.05–12.50)	0.5466	1.93 (0.06–17.35)	0.3552
Lymph node metastasis	No	ref		ref	
	Yes	1.72 (0.23–12.70)	0.9963	2.95 (0.22–31.75)	0.3320

## Data Availability

The data presented in this study are available on request from the corresponding author. The data are not publicly available due to data is being prepared for intellectual property approval.

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
