# Peer review of "The Novel Association of Early Apoptotic Circulating Tumor Cells with Treatment Outcomes in Breast Cancer Patients"

_ijms, 2022, doi:10.3390/ijms23169475_

Round 1

Reviewer 1 Report

Dear Authors:

Authors in this study examined the association of patient survival with sub-populations of CTCs with non-apoptotic, early apoptotic, end-point apoptotic, stem cell-like features by flow cytometry.

The review procedure can’t undergo fully because the content of last figure is missing.

One main concern is although authors focus on proportion of each subpopulations, I think total CTC number and how they correlate with survival should also be presented.

In figure 6 and 7, patient number is low and % of CTC is highly variable and this questions the viability/ reliability of statistical analysis

Figure 8 and 9 are very vague and not related to other data

Authors should check on scientific writing, some errors are spotted through the text. for example line31, 54, 75-76, 110, 114, etc

Author Response

I would like to thank you for careful and thorough reading of the manuscript and for the constructive suggestions, which help to improve the quality of this manuscript.

I suppose that some technical mistake take place with Figure 9, as I had downloaded the manuscript from the online platform and it's looking correct and contain all figures. I hope it will be looking correct after I will upload the manuscript with revisions.

Thank you for the suggestion to add data concerning to the correlation of total CTCs count with metastasis-free survival. No correlation of total CTC count with metastasis-free survival was found (p=0,4754). We have added this data in the manuscript.

We recognize that the sample size is small, and it is a serious limitation of our study. We considered it possible to carry out statistical processing of these data, realizing that they certainly require further validation. We suppose that the pattern revealed on such a small sample size will probably be confirmed in the validation.

Our study has shown that CD44+CD24- apoptotic CTCs (stem apoCTCs) associated with neoadjuvant chemotherapy response in breast cancer patients. Further, we clarified which exact subpopulations of apoCTCs (according to EpCam, cytokeratin 7/8, N-cadherin, CD133 and ALDH1 expression) associated with neoadjuvant chemotherapy response. On the figures 8 and 9 presented proportions of CD44+CD24- apoptotic CTCs with epithelial features - EpCam and cytokeratin 7/8 (CK7/8), mesenchymal feature (N-cadherin) and additional features of stemness (CD133 and ALDH1). We made legends of figures 8 and 9 more detailed and extended.

We revised the spelling and stylistic of the manuscript, furthermore the text was proofread by native-speaking person.

All edits in the manuscript are marked in green.

Reviewer 2 Report

The manuscript requires some extensive editing, both grammatical and scientific, before it can be published.

1  1.          First of all, English is not my first language and I cannot judge ultimately, but I feel that the manuscript will benefit from English proofreading. Some grammar or choice of words feel a little off to me. For example, in the Abstract lines 17 and 19, instead of “traits” of apoptosis would it be better to use “features” or “characteristics”? Line 15: maybe “focused on” instead of “dedicated on”? Line 114, 122: “and” instead of “и”. And so on, throughout the text.

22.       Line 122: was Annexin V used here, or anti-Annexin V antibody?

33.       Line 139-140: These cell lines are positive and negative controls for what?

44.       Please describe the gating strategy. How were CTCs defined?

55.       What was the percentage of CTCs in patients, compared to all non-CTC cells?

66.       In all figures, the legends are very brief. Please add more information, the figures are not self-explanatory now.

77.       Figure 1: needs more explanation. If I understand correctly it’s 1) non-apo CTCs, 2) apoCTCs, and 3) end-stage apoCTCs that make together 100% if we sum up each patient’s data? So each patient has three dots, in black, blue and greenish? I don’t think it’s correct to compare those three categories with Mann-Whitney (or some other) test. Comparison of apoptotic versus non-apoptotic cells percentages is not appropriate.

88.       Lines 205-209: How does prognostic value in Jansson et al. relates to prognostic value in this manuscript, in this case?

99.       Figure 4 C and D: do Grade 3 and Distant MTS groups have 1 patient each? The statistical test cannot be executed in this case, and it cannot be plotted on a diagram with a median shown. Same thing may be worrying at the other figures with n=2 or 3 patients per group only.

110.   Figures 3A, 4A, 6: how do you explain that, for example, non-apo CTCs decrease in one group compared to another, but apo-CTCs are not increased at the same time?

111.   Figure 7: Can we trust statistical difference when a group n=3?

112.   Figure 9: the bars or dots on the diagram are missing.

Author Response

I very much appreciated the critical and constructive comments on this manuscript that you made. We firmly believe that the comments and suggestions will greatly enhance the scientific value of the revised manuscript. All edits in the manuscript are marked in green.

  1. We made edits in the manuscript according to reviewer suggestions. We revised the spelling and stylistic of the manuscript, furthermore the text was proofread by native-speaking person.
  2. In our study we used Annexin V conjugated by FITC which can stain the phosphatidylserine on the outer cellular membrane when the cells initiate the apoptotic program. Combining with the staining of DNA in the cell nucleus with 7-Aminoactinomycin D (7-AAD) one can distinguishing viable cells from apoptotic cells and necrotic cells.
  3. In our study we used MCF‐7 cells as positive control and U937 cells as negative control for antibody specificity testing. Namely, MCF‐7 cells are EpCam-positive, CD45-negative. U937, vice versa, are EpCam-negative and CD45-positive.
  4. Gating strategy was as follow: using forward (FSC) and side scatter (SSC) gates debris was discriminate, doublets was also discriminate by plotting FSC area vs FSC height. Follow analysis included only CD45-negative cells, CTCs gated using quadrant-based scheme using EpCam and cytokeratin 7/8 to distinguish among the three CTCs subsets: EpCam+CK7/8-; EpCam-CK7/8+; EpCam-CK7/8-.
  5. In our study CTCs were extremely rare events in each CTC-positive patient and amount 0,01-0,02% of all blood cells.
  6. We made legends of figures 8 and 9 more detailed and extended.
  7. Figure 1 represented percentage of the apoCTC, non-apoCTC and end-stage apoCTC of distinct phenotypes: stem and non-stem (CD44+CD24-/CD44-CD24-) with simultaneous positive/negative expression of N-cadherin. So, CD44+CD24+ and CD44-CD24+ CTCs wasn’t studied. For this reason, the amount does not represent 100%. We added figure showing the percentages of non-apoCTCs, apoCTCs and end-stage apoCTCs in each breast cancer patient to the Supplementary. Indeed, three categories of CTCs – non-apoCTC, apoCTC end-stage apoCTCs are the dependent variables, so in that comparison Wilcoxon test was used.
  1. In our study no differences in apoCTCs proportion in untreated patients and patients treated with neoadjuvant chemotherapy with different molecular subtypes was detected (Figure 2). Jansson et al. also found no correlation between apoCTCc and molecular subtype while assessed of prognostic value of apoptosic CTCs in metastatic breast cancer patients. Corresponding edits made in the manuscript.
  2. We have modified the Figure 4, only data with significant differences in patients with different tumor size where the sample size was more than 3 had shown. Namely, in group of untreated patients there were 15 cases with T1 and 19 cases with T2. While in group of patients treated by NAC – 4 and 9 cases, correspondingly.
  3. Decrease of non-apo CTCs without simultaneous increase of apo-CTCs could be explained by the fact which comes from major limitation of our study. Namely, we assessed not all spectrum of phenotypes according to the CTCs markers that we used, but only distinct phenotypes: stem and non-stem (CD44+CD24-/CD44-CD24-) with simultaneous positive/negative expression of N-cadherin. Part of CTCs, for example CD44+CD24+ and CD44-CD24+ CTCs stay out of our study.
  4. We recognize that the sample size is small, and it is a serious limitation of our study. It is known that 2-year distant metastasis rate in breast cancer patients is 10-15%. In our study, which included 58 patients, metastases developed in 3 patients. We considered it possible to carry out statistical processing of these data, realizing that they certainly require further validation.
  5. We suppose that some technical mistake take place with Figure 9, as I had downloaded the manuscript from the online platform and it's looking correct and contain all figures. I hope it will be looking correct after I will upload the manuscript with revisions.

Reviewer 3 Report

A modern and relevant study was very upset by the carelessness of the preparation of the manuscript.

In particular, the abstract is blurred, concrete data is not provided

For some reason, the description of patients is included in the results...

technically, it's a simple job, but even the conditions for dividing blood into plasma and cells are not specified... It is very intriguing why the blood collected in EDTA should be incubated for an hour and a half at 37?

how many antibodies were added to each sample during cytometry? how were the samples aligned?

there are also a lot of questions about the figures. for example, what do the dashes in Figures 1-4, 6,7... mean? median?

Figure 5 shows the time in minutes? centuries?

Author Response

We are appreciated to the critical comments and will make every effort to improve our manuscript so that it meets the high requirements of the International Journal of Molecular Medicine.

We have added more own data obtained in current study to the Abstract.

The presentation of detailed data on the clinicopathological characteristics of patients in the Results section is widely practiced. Moreover, we have studied the articles of other authors published in IJMS and verified the correctness of such a presentation.

We used the whole blood separation technique without centrifugation which consists of passive settling of EDTA-treated whole blood to obtain white cell concentrate. This method is time-consuming but produces a more viable sample. To accelerate the process, we incubated blood samples at 37 degrees of Celsius.  

Each antibody was added according to the manufacturer instruction, namely, 5ml per test. The appropriate isotype control antibodies at the same concentration were added to the control sample.  It is not clear what exactly the reviewer means by the term "alignment" in relation to flow cytometry. Given that a multicolor panel was used, a compensation and FMO-control was performed.

Indeed, it is not correct to indicate median value in small groups, so we have modified the figures.  For example, Figure 4, only data with significant differences in patients with different tumor size where the sample size was more than 3 had shown.

Figure 5 demonstrates proportion (Y axis) of non-apoCTCs, apoCTCs and end-stage apoCTCs in breast cancer patient (X axis) with different responses to neoadjuvant chemotherapy.

Round 2

Reviewer 1 Report

Dear Authors,

I suggest authors to add the patient numbers in figure 2, 3, 5, and 6 when authors used bar plots instead of dot plots. In this way it is more clear for readers even with limited patient number.

Author Response

Dear Reviewer,

thank you for such a constructive suggestion that will really improve our manuscript and will make our data more clear to the readers. We have indicated information about the number of patients in the indicated figures.

Reviewer 2 Report

The authors introduced a lot of changes to the manuscript and it now looks much better. However I think some minor concerns still need to be addressed:

1) Author's response to comments 4 (gating strategy), 5 (total amount of CTCs), and 7 (study limitations) should be added to manuscript text.

2) The comparison at the Figure 1 still doesn't seem right. In my opinion, the Figure 1 is illustrative enough without any statistical analysis and can be left in its present view but without comparisons.

Author Response

Dear Reviewer,

thank you for all your hard work, we agree with your comments and have made the appropriate edits in the manuscript.

Reviewer 3 Report

The authors answered well, so the manuscript is suitable for publication.

Author Response

Dear Reviewer,

We are very grateful for the hard work you've done to improve our manuscript.